# Functional Trait Variation and Reverse Phenology in the Tropical Dry Forest Species *Bonellia nervosa*

**DOI:** 10.3390/plants14172659

**Published:** 2025-08-26

**Authors:** Ciara Duff, Bridget McBride, Gerardo Avalos

**Affiliations:** 1Environmental Studies Department, Mount Holyoke College, South Hadley, MA 01075, USA; ciaramarieduff@gmail.com (C.D.); mcbri23b@mtholyoke.edu (B.M.); 2Escuela de Biología, Universidad de Costa Rica, San José 11501-2060, Costa Rica; 3The School for Field Studies, Center for Ecological Resilience Studies, P.O. Box 506, West Boxford, MA 01885, USA

**Keywords:** diminishing returns, drought adaptation, leaf economics spectrum, phreatophytic character, seasonal dry forests

## Abstract

*Bonellia nervosa* is an understory tree with reverse phenology in tropical dry forests (TDFs), where seasonal water and temperature stress typically shape plant phenology and trait expression. This species is heliophytic and phreatophytic, relying on high light availability and deep-water access during the dry season. However, the role of dry-season light variation in influencing leaf traits of species with inverted phenology remains poorly understood. We examined how plant size, reproductive stage, and canopy structure influence trait variation in *B. nervosa* during the dry season. We measured plant height and diameter, reproductive status, and canopy structure using hemispherical photographs to estimate canopy openness, leaf area index, and transmitted light. Leaf structural traits included specific leaf area (SLA), thickness, water content, and stomatal density, while photochemical performance was assessed via chlorophyll fluorescence and rapid light curves. Principal component analysis and linear regression were used to examine trait–environment relationships. Photosynthetic efficiency was not affected by plant size or reproductive status. No strong trait correlations were observed for leaf water content and stomatal density. A negative relationship between canopy openness, transmitted light, and SLA indicates structural leaf adaptation to light conditions, with lower SLA values occurring under reduced light. In *B. nervosa*, leaf traits are driven more by light than by water availability during the dry season. This suggests that reverse phenology in phreatophytic species is functionally decoupled from seasonal water stress.

## 1. Introduction

Tropical dry forests (TDFs) are characterized by extreme seasonal fluctuations in precipitation, with up to 95% of annual rainfall occurring during the wet season in Costa Rica’s northwestern TDFs [1,2]. These prolonged dry periods have strongly influenced the evolution of plant structural and functional traits, which reduce competition for limited resources [3], resulting in a broad diversity of drought-adapted strategies and phenologies [4]. Precipitation seasonality and factors such as groundwater access and soil fertility shape ecosystem function by driving variation in key functional traits [5,6,7]. Species are positioned along a continuum of resource-use strategies that reflect varying degrees of drought tolerance, as determined by the variation in functional traits [3,8,9,10,11]. While water availability is widely recognized as the dominant environmental filter in TDFs [12,13,14], light distribution also plays a crucial role that is often underestimated [15]. Seasonal changes in canopy openness, leaf area index (LAI), and transmitted light generate sharp contrasts in understory light conditions, from the high-light environments of the dry season to the densely shaded understory typical of the wet season [16,17,18]. These seasonal shifts in light regimes further shape plant functional responses and should be integrated into a more comprehensive understanding of resource limitations and plant succession in TDFs.

*Bonellia nervosa* is a deciduous, heliophytic understory tree (i.e., a species adapted to full sunlight), frequently found in old secondary forest patches of TDFs [19]. *Bonellia nervosa* exhibits reverse phenology: it produces leaves, flowers, and fruits during the dry season (when most other deciduous species are leafless) and sheds its leaves at the onset of the rainy season [20,21,22]. This pattern is supported by a deep root system (phreatophytic character) that taps groundwater in the dry season [23]. By deploying a full crown during periods of reduced light competition in the dry season, *Bonellia nervosa* can take advantage of seasonal shifts in light availability [20,22].

Inverted leaf phenology makes *B. nervosa* an ideal candidate for expanding our understanding of phenological and functional strategies in TDFs, yet its exclusion from TDF trait-based analyses [9,24,25,26] reveals a significant gap in how functional diversity is represented in these ecosystems. Addressing this gap requires a framework that captures how plant traits respond to environmental gradients, for which functional ecology offers a robust foundation [27]. Within this framework, the leaf economics spectrum (LES) proposes that plant strategies fall along an acquisitive–conservative continuum, based on trade-offs between resource acquisition and leaf longevity [28,29]. Acquisitive species exhibit high photosynthetic rates, high specific leaf area (SLA), and short leaf lifespans (LL), whereas conservative species display the opposite traits. A related axis in TDFs is the evergreen–deciduous continuum, which reflects contrasting drought-coping strategies: evergreens typically possess deep roots, low SLA, and tough, long-lived leaves, while deciduous species generally have high SLA, short LLs, and shallow root systems [30]. *Bonellia nervosa* has the deep roots of evergreens but also the short leaf lifespan of deciduous species in tropical dry forests, showing that it does not fit the usual patterns; the analysis of its phenological behavior could help improve trait-based models in TDF ecology.

The LES literature has centered comparisons at the interspecific level [31,32], but growing evidence highlights the role of intraspecific variation in shaping structural and functional trait patterns [33,34,35,36,37]. Such variation may account for a significant portion of community trait diversity [36,38]. The diminishing returns hypothesis (DRH) identifies life stage as a key driver of trait variation within species. As plants increase in size, the proportion of photosynthetic to structural tissue declines, resulting in reduced resource-use efficiency, higher maintenance costs, and slower growth [34,39,40]. Thus, as plants grow, their resource use switches from an acquisitive to a conservative strategy. The DRH helps explain ontogenetic niche shifts and has been documented in multiple taxa [41,42,43].

Leaf-level photosynthetic performance, when considered alongside structural traits, may provide key insights into whether trait variation in *B. nervosa* aligns with the LES and DRH frameworks. Chlorophyll fluorescence techniques measure electron transport rates (ETRs) and energy fluxes, which serve as proxies to analyze changes in photochemical efficiency in response to different light conditions [44,45]. Rapid light curves (RLCs) reflect the saturation capacity of electron transport and may be used to assess the overall photosynthetic performance of plants [46]. These physiological metrics have been connected to LES traits in other species [47,48], but these relationships remain underexplored in *Bonellia nervosa*.

This study investigates two potential sources of structural and photochemical leaf trait variation in *B. nervosa*: ontogenetic development (as reflected by plant size and reproductive status) and microclimatic light conditions (captured by variation in canopy structure). We hypothesize the following:**Trait variation will correlate with plant size and reproductive status**: smaller, non-reproductive plants maximize light capture and growth by having thinner, larger leaves and a more acquisitive physiology (i.e., higher SLA, greater photosynthetic efficiency). As they grow larger and become reproductive, their strategy shifts toward a conservative and stress-tolerant strategy, developing traits that help them survive in a more variable and harsher environment (i.e., lower SLA, relatively lower photosynthetic efficiency), consistent with DRH.**Canopy openness will modulate trait expression (i.e., the light gradient hypothesis)**: in more open canopies, plants maximize light capture and show higher photochemical performance, while under more shaded canopies, traits become more conservative due to low light. This pattern would be especially pronounced in *B. nervosa*, whose inverted phenology and phreatophytic character mean light, not water, limits growth during the dry season. Therefore, individuals in more open canopies are expected to exhibit higher photochemical performance during this period.

By linking trait variation to plant size and changes in canopy structure, our study would improve the understanding of intraspecific plasticity and phenological specialization in TDF species. These insights provide a basis for predicting the responses of phreatophytic species with inverted phenology to climate variability and ecosystem disturbance.

## 2. Results

### 2.1. Definition of the Threshold for Reproductive Status

Our sample of 54 individuals ranged from 0.045 to 7.14 m in height and 0.15 cm to 27.20 cm in diameter. The Chapman–Richards model, using our combined data (this study and [18]), estimated the inflection point at 1.4 m, closely matching the height of the smallest individual with reproductive structures (1.0 m). Therefore, we decided to use 1.0 m as the threshold to distinguish reproductive (≥1 m) from non-reproductive (<1 m) individuals. While these categories are somewhat arbitrary, the model guided the definition of ontogenetic boundaries. The Chapman–Richards model simply identifies the point where the slope of the height–diameter relationship changes significantly. In the scatterplot (Figure 1), four individuals over 6 m deviate noticeably from the overall sigmoidal pattern. Following our criteria of reproductive individuals ≥ 1 m in height, 24.5% of our sample were non-reproductive, and the remaining 75.5% were reproductive.

Slenderness values were highly variable in small individuals, but this variation decreased with increasing plant size and eventually stabilized in the largest individuals. The negative relationship between slenderness and diameter indicates that as plants grow, they gain more in diameter relative to height, generating a robust, stocky architecture (Figure 2).

### 2.2. Variation in Canopy Structure and Functional Traits

Canopy openness ranged from 19.8% to 51.5%, total transmitted light from 23.2% to 69.7%, and LAI from 0.34 to 1.73, indicating considerable variation in canopy structure during the dry season across plants. However, reproductive and non-reproductive individuals experienced similar light environments (Figure 3). Most functional traits varied more in reproductive plants, but the overlap between reproductive and non-reproductive groups was considerable, except for height and diameter, which differed noticeably (Figure 3Q,R). Overall, both light conditions and trait variation were similar between reproductive and non-reproductive plants.

### 2.3. Correlation of Canopy Structure and Functional Traits

The correlation matrix revealed strong relationships within groups of traits, including variables related to canopy structure, leaf structure (e.g., LWC, SLA), RLC-derived traits, and plant size (height and diameter). Correlations among these trait groups were generally moderate (≥±0.30). On the other hand, stomatal density and leaf thickness showed weak correlations with the rest of the traits and were excluded from the PCA (Figure 4).

### 2.4. Principal Component Analysis

We performed a PCA using variables from the four trait groups, excluding those traits previously noted. We selected the first three components, which explained 64.5% of the variation. The first component (PC1, 33.8%) was primarily associated with canopy structure, with canopy openness and transmitted light loading positively and leaf area index (LAI) loading negatively (Table 1). The second component (PC2, 18.2%) was dominated by RLC parameters, *ETRs*, *ETRmax*, *β*, and Ek, all positively correlated. The third component (PC3, 12.6%) reflected variation in plant height, diameter, and leaf water content (LWC).

Based on the first two PCA components (51.94%), reproductive individuals showed a broader variation than non-reproductive individuals, which indicates greater phenotypic variability in relation to canopy structure (Figure 5 and Figure 6). Higher variation in reproductive plants was also the case for components 2 and 3 (Figure 6). When comparing changes in reproductive status along the first three principal components, significant differences were found only in the third component. Reproductive individuals scored higher on PC3, likely reflecting their larger size and greater LWC (F_1,51_ = 36.4, *p* < 0.0001, Figure 6).

### 2.5. Relationships Between Canopy Structure, Plant Size, SLA, and Photosynthetic Performance

We examined whether plant size (height and diameter) was related to canopy structure, summarized by the PC1 scores. Additionally, we assessed whether leaf structure (SLA) was associated with canopy structure. To test these relationships, we performed simple linear regressions between these variables and the PC1 scores (Figure 7). SLA decreased with increasing PC1 scores, indicating a positive association with leaf area index (LAI) and a negative association with canopy openness and transmitted light (Figure 5B and Figure 7C). In contrast, plant height and diameter showed no biologically meaningful relationships with PC1.

We also explored whether plant size and SLA were related to photosynthetic performance, as summarized by the second principal component (PC2), which captured variation in RLC parameters (Figure 5A). In this case, we found a weak but positive relationship between plant size and PC2 scores (Figure 7D,E), suggesting that as plants grow larger, their photosynthetic capacity slightly increases. SLA was not related to PC2. These patterns were also reflected in Figure 5.

### 2.6. Photosynthetic Efficiency

The RLC results showed similar characteristics of electron transport saturation in reproductive (N = 36) and non-reproductive (N = 17) individuals (Figure 8). The average saturation irradiance (Ek) was 220.29 µmol·m^−2^·s^−1^ for reproductive individuals and 212.24 µmol·m^−2^·s^−1^ for non-reproductive ones. Average ETRmax was slightly higher in reproductive (38.22) compared to non-reproductive individuals (35.22), while *α* was nearly identical between groups (0.17 vs. 0.16, respectively). These differences were statistically significant by reproductive status (two-way ANOVA, main effect of reproductive status: F_1,599_ = 6.6, *p* = 0.01). The interaction between reproductive status and PAR was non-significant (F_1,599_ = 0.44, *p* = 0.94), indicating that both groups exhibited curves of similar shape.

## 3. Discussion

### 3.1. Canopy Structure Drives Leaf Trait Variation

We found that the variation in leaf traits of *B. nervosa* was primarily driven by the local light environment rather than plant size or reproductive status. While SLA responded significantly to canopy structure, photochemical traits (Fv/Fm and RLC parameters) remained largely stable across sizes and reproductive stages. Fv/Fm values remained within the expected range for non-stressed plants, indicating no evidence of photoinhibition or drought stress [49,50]. The RLC parameters observed here were typical of unstressed, healthy plants [48,51]. Overall, the leaf cohort of *B. nervosa* was well adapted to the dry season radiation levels, with no evidence of photoinhibition. The decoupling between structural and physiological traits is particularly relevant for understanding the ecological consequences of a species with reverse phenology. SLA adjusts to canopy structure (e.g., higher SLA in low light, lower in high light, congruent with a conservative strategy), optimizing light capture efficiency. The negative relationship of SLA with canopy openness and transmitted light, alongside its positive correlation with LAI, indicates that, as the canopy closes, leaves tend to have lower SLA. This reflects structural adaptations where leaves become thicker and denser under shaded, low-light conditions in closed canopies.

*Bonellia nervosa* starts producing leaves before the onset of the dry season in a highly synchronous flush with a relatively rapid leaf expansion (mean synchrony of leaf production = 0.88 ± 0.06, N = 36 individuals, leaf expansion = 20.5 ± 5 days, N = 155 leaves; [22]), allowing a rapid structural adjustment to match the local light environment. Leaf bud production typically starts from late-September to mid-October [22,52] after the autumnal equinox (i.e., after a decrease in photoperiod and canopy cover), and leaf expansion is largely completed by late November, yielding nearly 70% of maximum crown area by the onset of the dry season. A combination of stored water, access to groundwater, and late rainy-season precipitation may facilitate the high synchrony in leaf production and the rapid leaf expansion. However, increasing canopy openness and decreasing photoperiod could trigger leaf bud production. Sánchez et al. [23] could not change leaf phenology through artificial irrigation, possibly because their treatments did not reach sufficient soil depth or the plants already had access to groundwater. In the study by Chaves & Avalos [22], direct light had a stronger effect on leaf production than plant size. The pronounced synchrony in leaf flushing and the rapid pace of leaf expansion may limit the timeframe for gradual post-expansion physiological acclimation. However, as observed here with SLA, this brief developmental window may still allow structural adjustments in response to local light conditions. These patterns suggest that leaf structure in *B. nervosa* is more strongly driven by pulses of light than by water availability, having adapted to a distinctive ecological niche that capitalizes on high understory light levels when competition for light is minimal.

Canopy cover varies markedly between the dry and wet seasons. Avalos et al. [18] reported that *B. nervosa* is typically found in canopy openings, averaging 14.68 ± 2.42%. However, in this study, canopy openness beneath *B. nervosa* during the dry season was substantially higher, averaging 31.67 ± 6.18% (range 20–51%), nearly double the rainy season value. By comparison, a mature forest patch at SRNP exhibited only 10.31% canopy openness in the dry season (pers. obs.). This considerable variation in canopy structure provides a strong selective pressure influencing leaf traits in *B. nervosa*, indicating that light availability is as critical for this species as water stress is for TDF deciduous and even evergreen species [22].

Leaf phenology is closely linked to the species’ reproductive schedule. Flower bud initiation occurs in mid-October and extends into late February, with peak flowering in January–February. Fruit development continues through May; in some cases, fruits remain on the plant for nearly two years [22]. A wide range of frugivores could disperse the orange, fleshy fruits, which average 12 ± 5 seeds (N = 37, [18]). The synchrony of leaf production contrasted with the lower flowering and fruiting synchrony [22], although flower initiation was influenced by direct light.

### 3.2. SLA Was More Responsive to Light Variation than Physiological Traits

SLA was the most responsive trait to variation in canopy structure, particularly transmitted light. This result supports predictions of the LES framework that individuals in more open canopies would exhibit lower SLA, a conservative strategy favoring thicker, denser leaves in high-light, water-limited environments [13,16,53,54,55,56]. The SLA values observed here (NR = 109 ± 13.4 cm^2^/g, R = 114 ± 20 cm^2^/g) fall under the average reported by Díaz et al. [56] regarding 10,486 plant species (138 cm^2^/g) but are comparable to SLA values of TDF species in general [57] and lean towards values observed for deciduous species (i.e., [25]). In TDFs, SLA reflects trade-offs between water conservation and photosynthetic capacity. SLA did not correlate with plant size or reproductive status, indicating that environmental heterogeneity, rather than ontogeny, dominates trait expression in *B. nervosa* during the dry season. SLA was more strongly related to canopy structure, showing negative loadings with canopy openness and transmitted light and positive loadings with LAI (Figure 5). This result highlights the value of integrating multiple canopy metrics when interpreting trait associations with canopy light conditions. Our findings are consistent with studies reporting that trait variation in TDFs is driven more by environmental factors than ontogeny [14,22] and that light variation within the dry season is sufficiently strong to affect SLA, in addition to expected changes in functional traits as caused by water stress and seasonal drought [16].

Local environmental conditions (e.g., soil moisture and community composition) drove trait variation more strongly than leaf habit alone (i.e., deciduous, semideciduous, or evergreen) in other TDF sites [9,14]. The inverse leaf phenology of *B. nervosa* results in highly synchronous leaf production, which likely contributed to the relative uniformity of the physiological trait variation across individuals, regardless of reproductive status or plant size. As a result, the leaf cohort produced each dry season is physiologically conservative, and SLA emerges as the key trait modulating responses to light variability. This variation was considerable, as indicated by the wide range of canopy openness, LAI, and transmitted light observed during the dry season. *Bonellia nervosa* appears to be well-adapted to old secondary forests [18], which maintain a structurally complex canopy, even during the dry season—an intermediate between that of young secondary stands and primary dry forests (pers. obs.).

### 3.3. Weak Support for the Diminishing Returns Hypothesis

Despite clear size-related differences in plant architecture—including a decline in slenderness with increasing trunk diameter—we found weak support for the diminishing returns hypothesis (DRH). Structural and physiological leaf traits remained largely unaffected by plant size or reproductive status, indicating that ontogenetic stage plays a minor role in shaping leaf-level function. These results may reflect the species’ strong phenological synchrony and rapid leaf expansion, which buffer leaf traits from gradual developmental effects predicted by the DRH [33,39]. Furthermore, the expectation that larger plants would become more conservative in resource use and exhibit lower SLA was not supported, likely due to a short effective functional leaf lifespan (about 5 months during the dry season) and the leaves being produced in a brief, synchronous pulse.

This pattern may also reflect greater investment in root development by larger individuals, allowing them to secure sufficient water resources and thereby dampening DRH effects. Meanwhile, small plants exhibit greater variation in morphology; this variation decreases as they grow larger and become more uniform. Reproductive individuals exhibited slightly higher ETRmax and Ek values. Still, these differences, although statistically significant, were small, and the overall shape of their light-response curves remained largely consistent across groups. Consequently, plant size and reproductive status have limited influence on photosynthetic performance, suggesting a relatively stable photochemical capacity across ontogeny. This lack of strong differences does not reflect a reduced investment in non-photosynthetic structures; instead, it highlights a different ecological strategy. The temporal niche that *B. nervosa* inhabits necessitates extensive deep root development to survive the dry season, demonstrating a phreatophytic adaptation to seasonally water-limited conditions [18]. Therefore, reverse phenology is closely tied to this specialized water-acquisition strategy and underscores a fundamental trade-off between rapid growth and survival under stress. This combination of reverse phenology and deep-water access likely buffers leaf-level traits from ontogenetic shifts, effectively masking size- or stage-related patterns that might otherwise emerge under more water-limited phenologies. Since young individuals must divert resources to roots to survive to reproductive age, they may not exhibit the same acquisitive growth strategy seen in seedlings of other species [22]. Therefore, *B. nervosa*’s inverted phenology may explain the lack of an observable “switch” from an acquisitive to a conservative growth strategy. In water-limited ecosystems, trade-offs between aboveground and belowground allocation—particularly between shoot and root investment—play a central role in shaping community functional diversity [58], highlighting how increased root allocation can constrain aboveground growth.

### 3.4. Adaptive Significance of Reverse Phenology and Trait Conservatism

The trait conservatism observed across reproductive and non-reproductive plants may be a product of *B. nervosa*’s reverse phenology, which enables the species to maximize light capture during the dry season. Deep rooting and synchronized leaf production may also constrain trait flexibility [19,23]. SLA varied more widely among reproductive individuals, suggesting greater plasticity at later stages. Such variation may reflect an increased capacity for light capture or phenotypic adjustment once water stress is mitigated by deeper root systems. The apparent environmental sensitivity of SLA, in contrast to physiological traits, underscores the role of structural investment in shaping functional responses to microhabitat conditions in TDFs [35,59].

### 3.5. Implications for Dry Forest Ecology and Conservation

Our findings reinforce the role of intraspecific trait variation in mediating species’ responses to environmental gradients in tropical dry forests (TDFs). In *B. nervosa*, relatively low SLA values and stable photochemical and morphological traits across size classes and reproductive stages point to a highly integrated response to the seasonal interplay of light and water availability. This apparent trait conservatism likely stems from two key factors: the species’ tightly synchronized leaf production and expansion during the dry season, and its access to deep water sources via a well-developed root system. Together, these adaptations reduce dependence on transient surface moisture and allow the species to exploit ample windows of high light availability with minimal competition. However, this specialization may also impose constraints. In the face of climate change—particularly increased variability in precipitation and more prolonged droughts—species like *B. nervosa*, which rely on stable access to groundwater and exhibit limited photochemical plasticity, may be less resilient. Habitat fragmentation could further restrict access to deep water and to old-growth secondary forests, the preferred habitats of this species, thus amplifying vulnerability. Understanding the seasonal dynamics of leaf traits, especially in seedlings and younger individuals, will be critical for assessing the species’ capacity to cope with a changing climate. Our findings suggest that restoration and conservation of mature secondary dry tropical forests, as well as areas with high groundwater availability in the TDF, could ensure the longevity of *B. nervosa* amidst the uncertain threats of climate change. Restoration of fragmented areas could provide additional support for this vulnerable ecosystem.

## 4. Materials and Methods

### 4.1. Study Site

The study was conducted along the “Indio Desnudo” trail in the Área de Conservación Guanacaste, Sector Santa Rosa, Guanacaste, Costa Rica (SRNP, 10°50′04″ N, 85°36′45″ W; 295 m.a.s.l.). The site is classified as a tropical premontane forest [60]. The area is characterized by an old secondary forest that has been regenerating for approximately 80 years [18]. The region receives an average annual rainfall of 1423 mm, with a pronounced dry season from December to April and a wet season from May to November. The yearly mean temperature is 25.7 °C, and the average relative humidity is 81% (SRNP climatic records). Fieldwork was conducted at the end of the dry season, from 4 to 5 March and 7–9 April 2025, during which we identified and measured all 54 visible *B. nervosa* individuals surrounding the trail, representing various reproductive stages and sizes.

### 4.2. Study Species

*Bonellia nervosa* is a member of the subfamily Theophrastoideae in the Primulaceae family [61]. The genus *Bonellia* comprises 22 species of dry biomes in Central America, the Greater Antilles, and northern and western South America [62]. *Bonellia nervosa* displays reverse phenology [20]. Although often described as a shrub, we observed individuals approaching 8 m in height, consistent with the definition of a small tree. This understory plant has coriaceous, simple, alternate leaves with a modified apex spine [63]. Leaf abscission happens in the mid-wet season in July–August, and leaf bud production begins in late September to mid-October [19]. The orange-red flowers bud one month after leaf production and are likely pollinated by hummingbirds or bees [20]. In the mid-dry season, they produce fruits one month after anthesis in February–March [22]. *Bonellia nervosa* is a phreatophytic species with deep tap and lateral roots compared to surrounding plants, allowing it to access water from the water table during the dry season [20].

### 4.3. Height-Diameter Relationship

We measured each plant’s height (H, in m) and diameter at breast height (DBH, in cm). For plants shorter than 1.5 m, the diameter was measured at three-quarters of the total height. In cases where individuals had multiple trunks emerging from a common base, we measured the DBH of each trunk and recorded the average. For individuals too tall to measure directly, we took photographs and used ImageJ software (version 1.54 p) to estimate height by comparing the plant to a reference object of known height.

We recorded the presence of reproductive structures for each individual. To account for cases where reproductive structures were absent in large individuals, we applied the Chapman–Richards growth model [64] to estimate a height-based inflection point indicative of reproductive maturity. For this analysis, we combined our dataset (N = 54) with that of Avalos et al. ([18], N = 33) to improve model robustness and expand the range of observations of reproductive structures. A shift in the height–diameter relationship was identified at 1.4 m in height (Figure 1), which may reflect a change in reproductive status. However, this threshold is a statistical estimate rather than a concrete biological cutoff. Since we observed a reproductive individual as short as 1.0 m, we adopted this height as the lower threshold for defining an individual as reproductive. However, the model identified a threshold consistent with field observations and allowed us to impute the reproductive status of a few individuals with no reproductive structures that fell within this range. We used this classification in subsequent analyses to compare traits between reproductive and non-reproductive individuals. The Chapman–Richards model also characterizes the relationship between plant height and stem diameter. To further explore this aspect, we analyzed the slenderness ratio (height/diameter) across individuals of varying sizes. Because height must increase while maintaining adequate mechanical support, slenderness in woody understory species can reflect architectural strategies for increased vertical growth under light-limited conditions. High slenderness ratios may indicate a preference for vertical elongation with low investment in radial growth, which is advantageous in a shaded understory.

### 4.4. Measurement of Leaf Structural Traits

To evaluate structural differences between reproductive and non-reproductive individuals, we collected five mature, fully expanded, and healthy leaves from each plant (two in the case of seedlings), randomly selected from the external portion of the crown. We averaged their measurements and stored the leaves in separate paper envelopes. We then measured leaf fresh weight, thickness, area, and stomatal density. We used a Scout Pro OHAUS Portable Electronic Balance scale (OHAUS Corporation, Parsippany, NJ, USA) to measure leaf fresh weight in g, and a Ubante micrometer with a resolution of 0.01 mm to measure leaf thickness in mm. We photographed the leaves and used ImageJ to measure their area in cm^2^. We determined leaf water content (LWC) by drying leaf samples in an oven at 50 °C for 24 h, or until reaching a constant weight, measuring each leaf’s dry weight and dividing it by its fresh weight. We calculated SLA (cm^2^/g) by dividing leaf area by leaf dry mass. To determine stomatal density, we applied a thin layer of clear nail polish on the underside of one leaf from each plant. Once dry, we peeled off the nail polish and placed it under a microscope to count the number of stomata within the field of view on a 400× objective. We recorded the number of stomata for each leaf per 0.159 mm^2^ (the area visible under 400× magnification).

### 4.5. Photosynthetic Efficiency Analysis

We used a Hansatech Instruments Handy PEA+ to measure the continuous excitation chlorophyll fluorescence emissions (Hansatech Instruments Ltd., Pentney, UK) and obtain parameters to describe the photosynthetic efficiency of each leaf sample. We measured rapid light curves (RLCs) by attaching a leaf clip to one leaf per individual and then positioning the light-emitting probe onto the clip. Actinic light was applied in 12 incremental steps (0, 10, 25, 50, 100, 175, 250, 350, 500, 750, 1200, and 1750 µmol·m^−2^·s^−1^), with each step lasting 10 s. In the resulting light response curves, we calculated key parameters, including the maximum electron transport rate (ETRmax), the light saturation coefficient (Ek), the initial slope under low light (α), and the rate of decline in photosynthetic electron transport after the curve reaches its maximum electron transport rate (β). These values were derived using the rapid light curve fitting function:$$ETR\;=\;ETRmax\;\cdot \;(1\;-\;e^{\left(-\alpha \cdot \frac{Par}{ETRmax}\right)})\cdot e^{\left(-\beta \cdot \frac{Par}{ETRmax}\right)}$$
with Ek = $\frac{ETRmax}{\alpha }$, following the formulation by [65]).

We additionally measured Fv/Fm, which represents the maximum quantum efficiency of Photosystem II and indicates the plant’s potential for photosynthesis (with non-stressed plants typically showing values between 0.79 and 0.84 [46]). To quantify Fv/Fm, we attached one leaf clip to five leaves of each adult individual and one clip to each seedling with fewer than five leaves, keeping the aluminum strip closed to induce darkness. We waited 10 min to allow the leaves to transition to a state of dark acclimation. We then attached the light-emitting probe to each clip and conducted the Fv/Fm measurement protocol on the Handy PEA+ device. The 10 min dark-acclimation period was determined following the manufacturer’s recommendations, and our preliminary tests showed parameter stabilization at this duration. Measurements were taken between 08:00 and 15:00 h, with a midday break, as our preliminary trials indicated no significant differences between morning and afternoon values under our experimental conditions. Following the methodology of Ralph & Gademann [44] we modeled RLCs in R Statistical Software version 4.5.0 [66] and packages minpack.lm [67] ggplot2 [68], and tidyr [69]. We modeled the RLCs for reproductive and non-reproductive individuals to compare electron transport at saturation between life stages.

### 4.6. Canopy Structure Analysis

We took hemispherical photos approximately 1.5 m above the ground, directly above or next to each plant. We used a Nikon Coolpix 5000 camera (Nikon Corporation, Tokyo, Japan) with a fisheye lens, mounted it on a tripod, and aligned each shot with the geographic North using a compass. Photographs were taken after dawn, near dusk, or under overcast skies to prevent glare from obscuring the images. We used Gap Light Analyzer software (GLA, version 2.0) to measure the percent canopy openness, leaf area index (LAI, measured at zenith angles of 0°–60°-LAI 4 and 0° to 75°-LAI 5), and percent transmitted diffuse, direct, and total light percent surrounding each plant. LAI represents the amount of leaf area per unit of surface area, which we used to measure forest structure and light availability [70].

### 4.7. Principal Component Analysis

To understand how size (DBH and H), leaf structure (thickness, water content, stomatal density, and SLA), photosynthetic performance (Fv/Fm, *ETR*, *ETRmax*, *α*, *β*, and Ek), and canopy structure (canopy openness, LAI 4 and 5, and transmitted light) are related, we performed a principal component analysis (PCA). We selected the first four components, summarizing 66.9% of the variation.

### 4.8. Regression Between Size and Canopy Structure (Predictor Variables) and Morphological and Fluorescence Variables (Response Variables)

We used linear regression models to examine the relationship between plant size, leaf structure, photosynthetic efficiency, and canopy structure parameters. To test our hypotheses that plant size and canopy structure affect structural and photochemical trait variation in *B. nervosa*, we conducted linear regressions between the principal components and each of the abovementioned traits. We additionally modeled a linear regression between SLA and height and between SLA and canopy openness. Data were inspected for normality, equality of variances, and the presence of outliers. We used R Statistical Software and JMP version 18.0 (JMP Statistical Discovery LLC, Cary, NC, USA) [71] for all statistical analyses.

## Figures and Tables

**Figure 1 plants-14-02659-f001:**
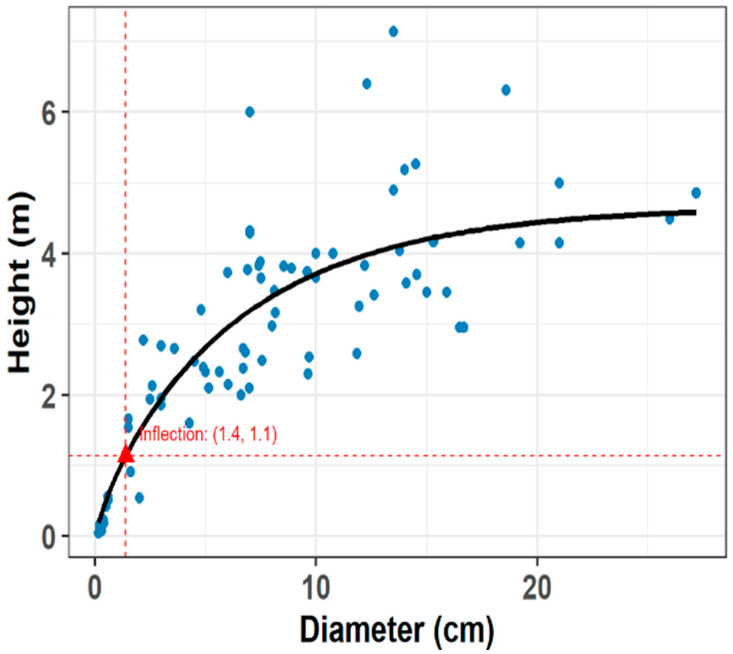
Trajectory of the Chapman–Richards model for *Bonellia nervosa* individuals in Santa Rosa National Park (N = 87 individuals). The model calculated an inflection point of 1.4 m in height and 1.1 cm in diameter.

**Figure 2 plants-14-02659-f002:**
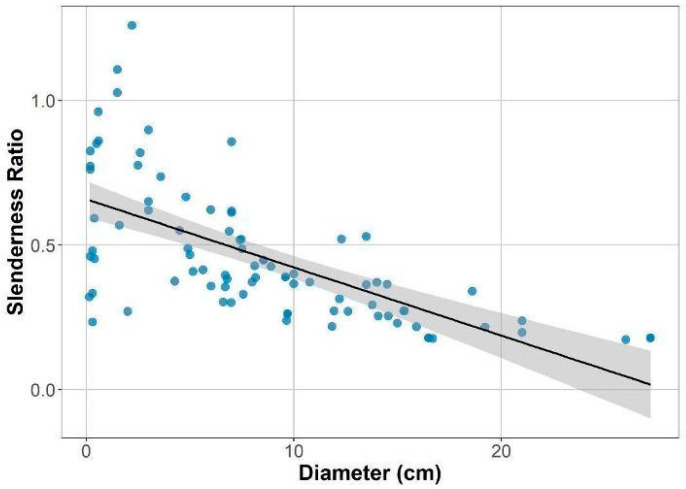
Slenderness ratio as a function of plant size (diameter) in *Bonellia nervosa* in Santa Rosa National Park. The relationship is approximated by a linear regression (slenderness = 0.66 − 0.02 ∗ diameter, R^2^ = 0.42, F_1,84_ = 62.98, *p* < 0.0001). The shaded area corresponds to the 95% confidence interval.

**Figure 3 plants-14-02659-f003:**
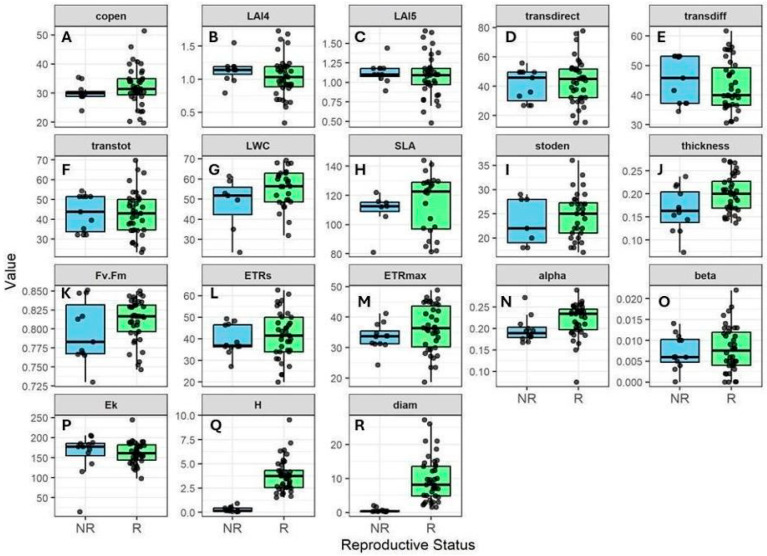
Variation in canopy structure and functional traits of reproductive (R) vs. non-reproductive (NR) *Bonellia nervosa* individuals in Santa Rosa National Park. The panels show: (**A**) canopy openness; (**B**) LAI4; (**C**) LAI5; (**D**) transmitted direct light; (**E**) transmitted diffuse light; (**F**) transmitted total light; (**G**) leaf water content; (**H**) specific leaf area; (**I**) stomatal density; (**J**) leaf thickness; (**K**) Fv/Fm; (**L**) electron transport rate; (**M**) maximum electron transport rate; (**N**) alpha (photosynthetic rate in light-limited region of rapid light curve); (**O**) beta (the ratio of light absorbed by Photosystem II); (**P**) Ek (minimum saturating irradiance); (**Q**) height; (**R**) diameter. Trait abbreviations follow Table 1.

**Figure 4 plants-14-02659-f004:**
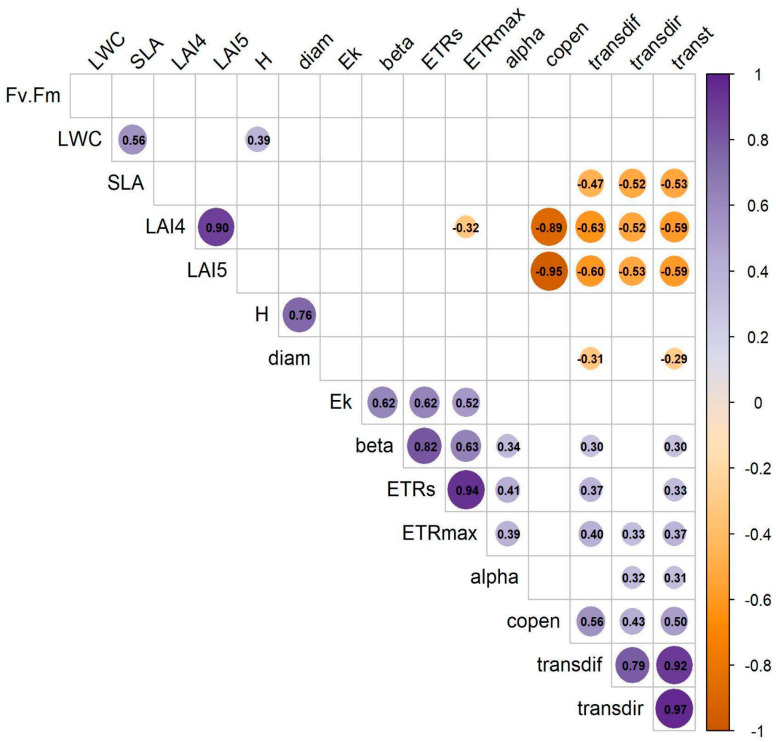
Correlation matrix of functional traits in *Bonellia nervosa* in Santa Rosa National Park. Trait abbreviations follow Table 1.

**Figure 5 plants-14-02659-f005:**
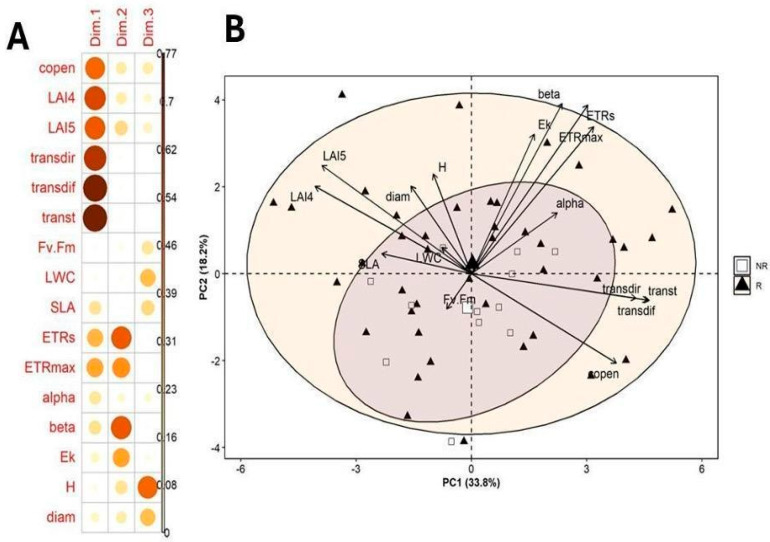
(**A**) Leverage of functional traits on the first three principal components, and (**B**) spatial distribution of reproductive and non-reproductive individuals in the space defined by the first two principal components in *Bonellia nervosa* in Santa Rosa National Park.

**Figure 6 plants-14-02659-f006:**
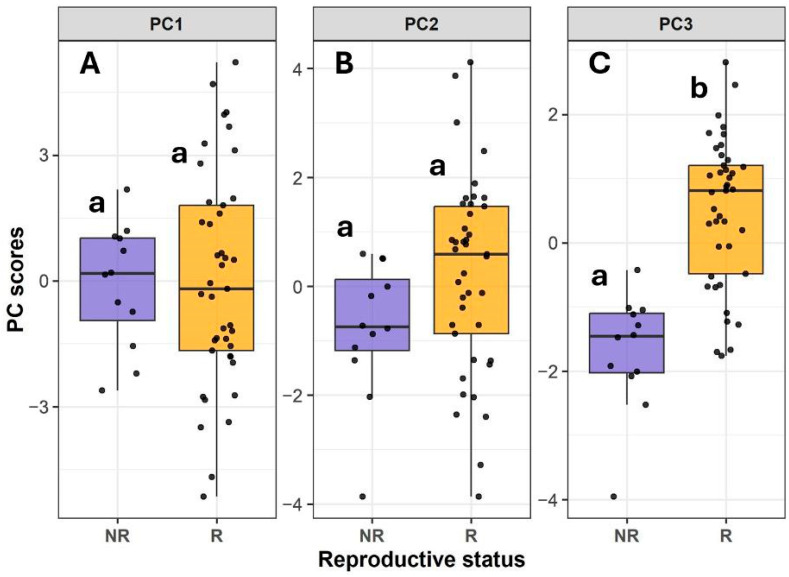
Variation in the first three principal components relative to reproductive status in *Bonellia nervosa* in Santa Rosa National Park: (**A**) Principal Component 1 (PC1) for reproductive (R) and non-reproductive (NR) plants; (**B**) Principal Component 2 (PC2) for reproductive (R) and non-reproductive (NR) plants; (**C**) Principal Component 3 (PC3) for reproductive (R) and non-reproductive (NR) plants. In the figure, a and b indicate significant differences.

**Figure 7 plants-14-02659-f007:**
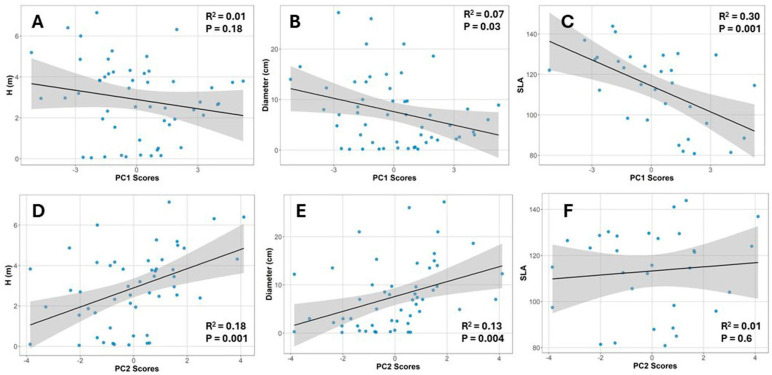
Relationships in *Bonellia nervosa* in Santa Rosa National Park: (**A**) plant height vs. canopy structure PC1; (**B**) plant diameter vs. canopy structure PC1; (**C**) specific leaf area (SLA) vs. canopy structure PC1; (**D**) plant height vs. rapid light curve parameters PC2; (**E**) plant diameter vs. rapid light curve parameters PC2; (**F**) specific leaf area (SLA) vs. rapid light curve parameters PC2.

**Figure 8 plants-14-02659-f008:**
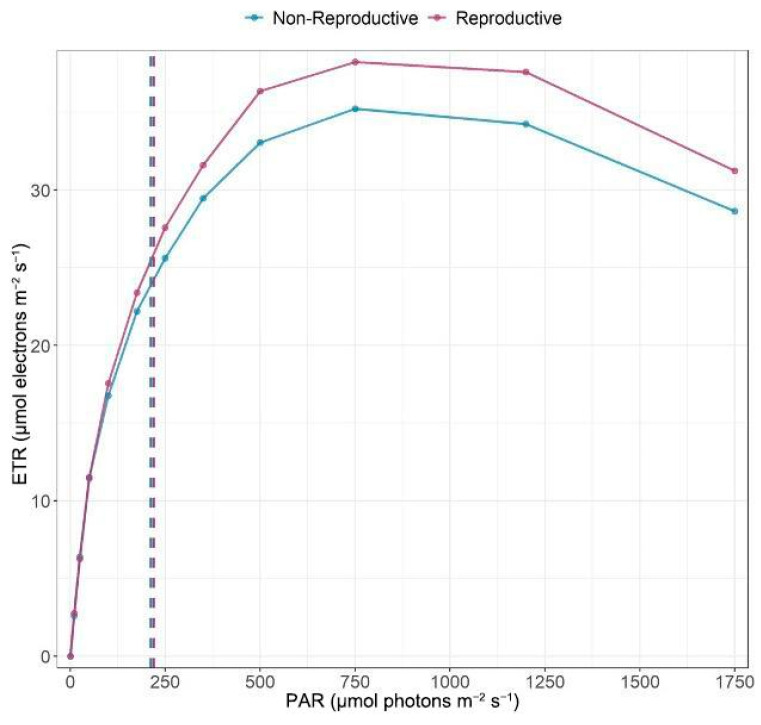
Average values of the rapid light curves of 36 reproductive and 17 non-reproductive individuals of *Bonellia nervosa* at Santa Rosa National Park, where electron transport rate (ETR) is plotted against photosynthetically active radiation (PAR). The dotted line corresponds to the parameter Ek, which is the light intensity at which the ETR transitions from a linear increase to saturation.

**Table 1 plants-14-02659-t001:** Eigenvalues and eigenvectors of the principal component analysis of canopy structure variables and functional traits in *Bonellia nervosa* in Santa Rosa National Park. The abbreviation for the functional trait and the unit of measurement are indicated in parentheses, if applicable. Bolded values indicate traits with strong associations with the principal component.

Component	Eigenvalue	Percent	Cum Percent
1	5.40	33.75	33.75
2	2.91	18.18	51.94
3	2.00	12.56	64.50
**Trait**	**PC1**	**PC2**	**PC3**
Canopy openness (copen, %)	**0.71**	−0.39	0.38
LAI4	**−0.77**	0.38	−0.29
LAI5	**−0.74**	0.47	−0.33
Transmitted direct light (transdir, µmol·m^−2^·s^−1^)	**0.81**	−0.10	−0.01
Transmitted diffuse light (transdif, µmol·m^−2^·s^−1^)	**0.87**	−0.11	−0.03
Transmitted total light (transt, µmol·m^−2^·s^−1^)	**0.88**	−0.11	−0.02
Fv/Fm	−0.12	−0.15	0.43
Leaf water content (LWC)	−0.14	0.11	**0.55**
Specific leaf area (SLA, cm^2^/g)	−0.44	0.08	0.48
Electron transport rate (ETR, µmol electrons m^−2^ s^−1^)	0.58	**0.73**	−0.05
Maximum electron transport rate (ETRmax, µmol electrons m^−2^ s^−1^)	0.60	**0.64**	−0.04
alpha	0.42	0.26	0.26
beta	0.45	**0.74**	−0.08
Ek (µmol photons m^−2^ s^−1^)	0.31	**0.61**	−0.17
Height (H, m)	−0.18	0.43	**0.72**
Diameter (diam, cm)	−0.30	0.38	**0.55**

## Data Availability

The data supporting the findings of this study are openly available in the Mendeley Data repository at https://data.mendeley.com/datasets/24hsbhw6wc/1 (accessed on 7 July 2025).

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
