# Peer review of "Functional Trait Variation and Reverse Phenology in the Tropical Dry Forest Species Bonellia nervosa"

_plants, 2025, doi:10.3390/plants14172659_

Round 1
Reviewer 1 Report
Comments and Suggestions for Authors
Comments to the Author
The article presents an important topic about functional trait variation and reverse phenology in the tropical dry forest species Bonellia nervosa. Overall, manuscript is well-written, but authors need to address most of the concerns raised about the experimental methodology. Moreover, pleases address minor issues in the manuscript based on the following comments.
- Line 16: The used terms “heliophytic” and “phreatophytic” may be hard to follow for general scientific audience. Consider simplifying them or adding brief definitions in brackets.
- Line 27-28: Its suggested to rephrase for clarity like this “Canopy openness was positively correlated with SLA, indicating structural adaptation to light conditions.”
- Line 28-30: Strong concluding idea but awkward wording. You can rephrase it like “in B. nervosa, leaf traits are driven more by light than water availability during the dry season.” You claim that “how reverse phenology in phreatophytic species is functionally decoupled from seasonal water stress.” Do your findings directly support it?
- Line 123: The section numbering and headings in the results/discussion are a bit confusing. For example, “2. Results” is immediately followed by a subsection labeled “3.1. Later, the Discussion is labeled as section 4, with subsections 4.1–4.5. Check it carefully.
- Line 297-300: There is a clarity issue regarding the relationship between SLA and canopy openness. The abstract states “SLA increased significantly with canopy openness, indicating structural acclimation to light,” whereas, here In the discussion, the authors note that in more open canopies one would expect lower SLA. Ensure the abstract accurately reflects the findings of the study.
- Line 429-430: In results section Line 125, authors claim the sample size of 54 individuals whereas here discussed that “we combined our dataset (N = 429) with that of Avalos et al. (N = 33) to improve model robustness.” The authors should clarify why one individual was excluded from the analysis.
- Line 450-451: The authors collected five leaves per plant to measure structural traits, but it’s unclear how these leaves were selected. Were they fully expanded, sun-exposed leaves from the upper canopy, or randomly chosen including shaded interior leaves? Leaf traits can vary with position and light exposure, so inconsistent sampling could introduce bias. Please describe the selection protocol e.g., same canopy position, healthy mature leaves to strengthen the methodology.
- Line 492-495: The hemispherical photography method for canopy openness may not capture conditions for taller individuals. Photos were taken ~1.5 m above ground “above or next to each plant,” which is suitable for short plants but for taller trees this height is below their canopies. The authors should clarify how they handled canopy images for taller trees. Ensuring the measurement reflects the light environment at the top of each plant is important for accuracy.
- Line 484: Need more clarification for chlorophyll fluorescence measurements. The authors claim “We waited 10 min to allow the leaves to transition to a state of dark acclimation” that’s relatively short. Many protocols have recommended at least 20-30 minutes for dark acclimation of leaves. Additionally, clarify the time of day when fluorescence measurements were taken.
Author Response
Reviewer 1
The article presents an important topic about functional trait variation and reverse phenology in the tropical dry forest species Bonellia nervosa. Overall, manuscript is well-written, but authors need to address most of the concerns raised about the experimental methodology. Moreover, pleases address minor issues in the manuscript based on the following comments.
- Line 16: The used terms “heliophytic” and “phreatophytic” may be hard to follow for general scientific audience. Consider simplifying them or adding brief definitions in brackets. RESPONSE: We added a definition for these two terms in parentheses the first time they are mentioned in the main text: "heliophytic (plants adapted to full sunlight)," lines 57-58. The "phreatophytic” term is defined in lines 61-62. We believe there is no room in the abstract to introduce these definitions without making the abstract unnecessarily wordy. In addition, the audience familiar with the phenology of tropical dry forest species must know these terms.
- Line 27-28: Its suggested to rephrase for clarity like this “Canopy openness was positively correlated with SLA, indicating structural adaptation to light conditions.” RESPONSE: This sentence was rephrased (lines 29-31) “A negative relationship between canopy openness, transmitted light, and SLA indicates structural leaf adaptation to light conditions, with lower SLA values occurring under reduced light.”
- Line 28-30: Strong concluding idea but awkward wording. You can rephrase it like “in B. nervosa, leaf traits are driven more by light than water availability during the dry season.” You claim that “how reverse phenology in phreatophytic species is functionally decoupled from seasonal water stress.” Do your findings directly support it? RESPONSE: We changed the wording, so it now reads (lines 31-33) “In B. nervosa, leaf traits are driven more by light than water availability during the dry season. This suggests that reverse phenology in phreatophytic species is functionally decoupled from seasonal water stress.” The ability of B. nervosa to access groundwater likely reduces its dependence on seasonal water availability, allowing leaf traits to respond primarily to variations in light during the dry season. This explanation aligns well with our observations and current understanding of phreatophytic species, making it a plausible functional interpretation of the data.
- Line 123: The section numbering and headings in the results/discussion are a bit confusing. For example, “2. Results” is immediately followed by a subsection labeled “3.1. Later, the Discussion is labeled as section 4, with subsections 4.1–4.5. Check it carefully. RESPONSE: We thank the reviewer for spotting this inconsistency. We also corrected the numbering and sequence of the figures and their correct reference in the main text.
- Lines 297-300: There is a clarity issue regarding the relationship between SLA and canopy openness. The abstract states, “SLA increased significantly with canopy openness, indicating structural acclimation to light,” whereas, here In the discussion, the authors note that in more open canopies one would expect lower SLA. Ensure the abstract accurately reflects the findings of the study. RESPONSE: In lines 260-262, we indicate that the lower SLA in high light and the higher SLA in low light correspond to a conservative leaf economics spectrum strategy. In lines 262-266, we included further clarification, adding this sentence: “The negative relationship of SLA with canopy openness and transmitted light, alongside its positive correlation with LAI, indicates that as the canopy closes, leaves tend to have lower SLA. The decrease in SLA reflects structural adaptations where leaves become thicker and denser under shaded, low-light conditions in closed canopies.” Our PCA results are consistent with these plasticity responses, where decreasing SLA with canopy closure is a typical adaptation to reduced light. Now in the Abstract (lines 28-31) it reads, “The negative relationship between canopy openness, transmitted light, and SLA indicates structural leaf adaptation to light conditions, with lower SLA values occurring under reduced light”. This makes consistent our conclusion in the Discussion with the summary of the findings in the Abstract.
- Line 429-430: In results section Line 125, authors claim the sample size of 54 individuals whereas here discussed that “we combined our dataset (N = 429) with that of Avalos et al. (N = 33) to improve model robustness.” The authors should clarify why one individual was excluded from the analysis. RESPONSE: Somehow the reviewer had the summation wrong. In this study, we measured 54 plants. In the Avalos et al. (2005) study, there were 33 plants measured. Thus, we had a total of 87 plants included in the combined dataset. This number is indicated in the legend of Figure 1. However, in line 449, there is an inconsistency indicating that this dataset had 53 observations; it had 54. We made the correction. Stomatal density and leaf thickness were excluded due to very low correlations as indicated in lines 173-175.
- Line 450-451: The authors collected five leaves per plant to measure structural traits, but it’s unclear how these leaves were selected. Were they fully expanded, sun-exposed leaves from the upper canopy, or randomly chosen including shaded interior leaves? Leaf traits can vary with position and light exposure, so inconsistent sampling could introduce bias. Please describe the selection protocol e.g., same canopy position, healthy mature leaves to strengthen the methodology. RESPONSE: We appreciate the reviewer’s observation and agree that clarifying the leaf selection protocol strengthens the methodology description. In our study, we collected mature, fully expanded, and healthy leaves located in the external portion of the crown within hand’s reach. Leaves were selected randomly from this accessible crown zone to ensure consistency among individuals while avoiding damaged or immature leaves. Lines 468-471 now read, “To evaluate structural differences between reproductive and non-reproductive individuals, we collected five mature, fully expanded, and healthy leaves from each plant (two in the case of seedlings), randomly selected from the external portion of the crown. We averaged their measurements and stored the leaves in separate paper envelopes.”
- Line 492-495: The hemispherical photography method for canopy openness may not capture conditions for taller individuals. Photos were taken ~1.5 m above ground “above or next to each plant,” which is suitable for short plants but for taller trees this height is below their canopies. The authors should clarify how they handled canopy images for taller trees. Ensuring the measurement reflects the light environment at the top of each plant is important for accuracy. RESPONSE: We appreciate the reviewer’s insightful remark regarding the positioning of the camera for taking hemispherical photographs for canopy openness measurements. Our approach follows established protocols for hemispherical photography (e.g., Chazdon and Field 1987, Rich 1990; Frazer et al. 1999), in which images are typically taken at a standardized height—here, 1.5 m above the ground—either above or next to each plant. This standardization is essential to maintain comparability across all sampling sites and individuals. While we acknowledge that this height does not replicate the exact light environment at the top of taller individuals, adjusting the camera height to match each plant’s canopy would introduce methodological inconsistencies and confound comparisons. Instead, our measurements represent the understory or mid-canopy light conditions experienced by leaves accessible within reach, which was the relevant scale for our structural trait measurements. This approach ensures robust, comparable data across the full range of plant sizes in our study.
References:
Chazdon, R. L., & Field, C. B. (1987). Photographic estimation of photosynthetically active radiation: evaluation of a computerized technique. Oecologia, 73(4), 525-532.
Rich, P. M. (1990). Characterizing plant canopies with hemispherical photographs. Remote sensing reviews, 5(1), 13-29.
Frazer, G. W., Canham, C. D., & Lertzman, K. P. (1999). Gap Light Analyzer (GLA), Version 2.0: Imaging software to extract canopy structure and gap light transmission indices from true-colour fisheye photographs, users manual and program documentation. Simon Fraser University, Burnaby, British Columbia, and the Institute of Ecosystem Studies, Millbrook, New York, 36.
- Line 484: Need more clarification for chlorophyll fluorescence measurements. The authors claim “We waited 10 min to allow the leaves to transition to a state of dark acclimation” that’s relatively short. Many protocols have recommended at least 20-30 minutes for dark acclimation of leaves. Additionally, clarify the time of day when fluorescence measurements were taken. RESPONSE: We appreciate the reviewer’s comment regarding the dark-acclimation period for chlorophyll fluorescence measurements. Our protocol followed the manufacturer’s recommendations in the Hansatech Instruments Pocket PEA manual, which specifies that a 10-minute dark-acclimation period is adequate for obtaining reliable maximum quantum yield (Fv/Fm) values in non-stressed leaves. In addition, we took preliminary data, which showed that fluorescence parameters stabilized at 10 min of dark acclimation in our species and experimental conditions. Several studies using the same instrument have adopted similar dark-acclimation durations in both tropical and temperate plant species (e.g., Maxwell & Johnson 2000). While longer acclimation times (15–30 minutes) are used in some protocols, these are generally applied in contexts where leaves are under stress or where photoinhibition is suspected. Regarding the possible diurnal variation in chlorophyll fluorescence measurements, we conducted preliminary tests measuring fluorescence both in the early morning (around 8 a.m.) and late afternoon (around 3 p.m.), with a break at noon. Our results showed no significant differences in fluorescence responses between morning and afternoon measurements under our experimental conditions. This suggests that the fluorescence parameters we report are stable throughout the period measured. We made these clarifications in the Methods in the lines 506-511, which read, “The 10-min dark-acclimation period was determined following the manufacturer's recommendations, and our preliminary tests showed parameter stabilization at this duration. Measurements were taken between 08:00 and 15:00 h, with a midday break, as our preliminary trials indicated no significant differences between morning and afternoon values under our experimental conditions.”
Reference: Maxwell K, Johnson GN (2000) Chlorophyll fluorescence—a practical guide. J Exp Bot 51:659–668. https://doi.org/10.1093/jexbot/51.345.659

Reviewer 2 Report
Comments and Suggestions for Authors
Dear authors, I have thoroughly and thoroughly reviewed your manuscript. However, I think it is important to focus on reviewing and addressing the following observations which I recommend:
Introduction
I believe it is necessary to include specific data on how seasonal changes in light and water availability affect the reverse phenology of B. nervosa compared to other dry forest species.
The authors should include previous studies that have documented reverse phenology in other TDF species to better contextualise the uniqueness of B. nervosa
It is important to mention something about intraspecific variation in functional ecology studies, as this aspect is often underestimated in favour of interspecific comparisons.
Provide examples of how phenotypic plasticity in response to light could confer competitive advantages in seasonal environments.
Materials and methods
Please explain the criteria for selecting the 54 individuals, including how you ensured you covered a representative range of sizes and reproductive stages.
Why was the Chapman-Richards model used to define reproductive status and discuss possible biases in this classification?
Discussion
Perhaps it would be necessary to pose the following formulation: Could reverse phenology and access to groundwater mask ontogenetic effects?
If possible, it would be necessary to compare these results with studies in other TDF species where HRD has been observed, highlighting the uniqueness of B. nervosa.
Dear authors, the dependence of B. nervosa on dry season light and groundwater could make it vulnerable to changes in precipitation regimes or habitat fragmentation.
Conservation recommendations, such as protection of areas with high groundwater availability or restoration of mature secondary forests where this species thrives, would need to be proposed.
Author Response
REVIEWER 2
Comments and Suggestions for Authors
Dear authors, I have thoroughly and thoroughly reviewed your manuscript. However, I think it is important to focus on reviewing and addressing the following observations which I recommend:
Introduction
I believe it is necessary to include specific data on how seasonal changes in light and water availability affect the reverse phenology of B. nervosa compared to other dry forest species. RESPONSE: The only reference linking Bonellia nervosa to changes in light availability, Chaves and Avalos (2008), has already been discussed in our manuscript. Sánchez et al. (2020) address a different topic, namely the manipulation of groundwater to influence leaf phenology, but their results were inconclusive. No other species with reverse phenology is known from tropical dry forests in the Americas. Chaves and Avalos (2008) demonstrated that direct light had a stronger effect on leaf production than plant size. Our results corroborate these findings using a larger sample size and additional light parameters, such as transmitted light and canopy openness. In our discussion, we note that studies of tropical dry forest phenology have overlooked this unusual species and that such analyses are incomplete without considering the ecological and evolutionary drivers of inverted phenology. We have referenced pertinent literature and believe the manuscript fully addresses this point.
References
Chaves, O. M., & Avalos, G. (2008). Do seasonal changes in light availability influence the inverse leafing phenology of the neotropical dry forest understory shrub Bonellia nervosa (Theophrastaceae)?. Revista de Biología Tropical, 56(1), 257-268.
Sánchez, O., Quesada, M., Dirzo, R., & Schlichting, C. D. (2020). A field experiment to determine the effect of dry-season irrigation on vegetative and reproductive traits in the wet-deciduous tree Bonellia nervosa. Journal of Tropical Ecology, 36(1), 29-35.
The authors should include previous studies that have documented reverse phenology in other TDF species to better contextualise the uniqueness of B. nervosa. RESPONSE: As mentioned above, there is no other known Neotropical species with reverse phenology, and we have made this point explicit and substantiated it with pertinent literature.
It is important to mention something about intraspecific variation in functional ecology studies, as this aspect is often underestimated in favour of interspecific comparisons. RESPONSE: This point is already covered in the manuscript (Lines 81-90), where we discuss its role in shaping structural and functional trait patterns, supported by relevant literature [33–37] as well as hypotheses such as the LES and the DRH. In those lines, we specifically note that intraspecific variation can account for a significant portion of community trait diversity [36,38] and frame this within the context of DRH, which explains how ontogenetic changes in plant size drive shifts from acquisitive to conservative resource-use strategies [34,39,40]. We believe this section adequately addresses the reviewer’s concern.
Provide examples of how phenotypic plasticity in response to light could confer competitive advantages in seasonal environments.
RESPONSE: We believe we addressed this topic in the Discussion (Sections 3.1, 3.2, 3.4), where we provide specific examples of how phenotypic plasticity in response to light confers competitive advantages in B. nervosa. We show that SLA varies predictably with canopy structure (i.e., higher SLA in shaded understories to maximize light capture and lower SLA in more open canopies). We also discuss how B. nervosa’s reverse phenology enables it to fully deploy its crown during the dry season, when understory light availability is highest and competition is minimal. This seasonal synchronization, combined with structural adjustments in leaf traits, allows the species to exploit a temporal niche, maximize resource acquisition during high‐light conditions, and maintain performance under low‐light. These are concrete examples of phenotypic plasticity directly enhancing competitive ability in a strongly seasonal environment.
Materials and methods
Please explain the criteria for selecting the 54 individuals, including how you ensured you covered a representative range of sizes and reproductive stages. RESPONSE: We appreciate the reviewer’s attention to precision in defining the sample size of individuals. Lines 129-132 of the results section indicate the range in size and reproductive status of the individuals sampled. To better clarify the selection process, lines 418-421 now read: “Fieldwork was conducted at the end of the dry season, from March 4–5 and April 7–9, 2025, during which we identified and measured all 54 visible B. nervosa individuals surrounding the trail, representing various reproductive stages and sizes.” This sampling secured a wide range of sizes and reproductive stages, providing a set of individuals largely representative of the population structure in the field.
Why was the Chapman-Richards model used to define reproductive status and discuss possible biases in this classification?
RESPONSE: We used the Chapman–Richards model as an alternative, more objective method to estimate changes in the growth pattern associated with ontogenetic changes and then classify reproductive vs. non-reproductive individuals. This model is well-suited for describing ontogenetic changes in plants, including reproductive status, as a function of size or age. The model accommodates the non‐linear pattern in which reproductive probability is low at small sizes, increases rapidly near the onset of maturity, and then plateaus in larger individuals. By fitting this model to our data, we could estimate the size threshold at which B. nervosa transitions to reproductive status, which allowed us to integrate reproductive maturity into the interpretation of trait variation and phenological patterns. However, we recognize potential biases in this model, since it assumes a smooth and continuous increase in reproductive probability with size, which may not capture year‐to‐year variability in reproductive effort due to environmental conditions, resource availability, or individual history. The model was an additional criterion to differentiate between reproductive and non-reproductive individuals.
Discussion
Perhaps it would be necessary to pose the following formulation: Could reverse phenology and access to groundwater mask ontogenetic effects?
RESPONSE: We thank the reviewer for this question. We agree that reverse phenology combined with access to groundwater could mask ontogenetic effects. As discussed in sections 3.3 and 3.4, B. nervosa exhibits a highly synchronous, rapid leaf flush at the onset of the dry season, coupled with deep rooting that ensures water availability. This strategy buffers leaves from prolonged developmental constraints, limiting the influence of plant size or reproductive stage on leaf functional traits. To make this connection more explicit, we have added a clarifying sentence at the end of section 3.3, lines 363-366: “This combination of reverse phenology and deep-water access likely buffers leaf-level traits from ontogenetic shifts, effectively masking size- or stage-related patterns that might otherwise emerge under more water-limited phenologies.”
If possible, it would be necessary to compare these results with studies in other TDF species where HRD has been observed, highlighting the uniqueness of B. nervosa.
RESPONSE: We assumed the reviewer meant DRH and not HRD. To our knowledge, and after doing a thorough search of the literature, we have been unable to locate explicit evaluations of the ‘diminishing returns hypothesis’ in tropical dry forest species. Thus, our finding of weak DRH support in B. nervosa not only underscores the strong structural and physiological conservatism in this species but also highlights its distinctiveness within a trait dimension seldom explored in TDF contexts.
Dear authors, the dependence of B. nervosa on dry season light and groundwater could make it vulnerable to changes in precipitation regimes or habitat fragmentation.
RESPONSE: We agree with the reviewer but have addressed this point already in section 3.5 (Implications for dry forest ecology and conservation). There, we highlight how the species’ specialization to dry-season light capture and deep-water access, coupled with limited photochemical plasticity, may reduce its resilience to increased precipitation variability, prolonged droughts, and habitat fragmentation. We also discuss how these vulnerabilities could be mitigated through restoration and conservation of mature secondary forests and areas with high groundwater availability.
Conservation recommendations, such as protection of areas with high groundwater availability or restoration of mature secondary forests where this species thrives, would need to be proposed. RESPONSE: We thank the reviewer for proposing conservation recommendations benefitting the species and ecosystem. In addition to those outlined in section 3.5 of the discussion, we added lines 403-406, which read, “Our findings suggest that restoration and conservation of mature secondary dry tropical forests, as well as areas with high groundwater availability in the TDF, could ensure the longevity of B. nervosa amidst the uncertain threats of climate change. Restoration of fragmented areas could provide additional support for this vulnerable ecosystem.”

Reviewer 3 Report
Comments and Suggestions for Authors
In this manuscript, Ciara Duff and colleagues examined how plant size, reproductive stage, and canopy structure influence trait variation in Bonellia nervosa during the dry season. I have following comments:
1, For the Abstract section, more results like thickness, water content, and stomatal density should be included.
2, For the keywords, words appeared in the title should be removed from the list.
3, For the introduction section, main conclusion and practical interests should be stated in the last paragraph
4, For the Table 1, error values should be presented.
5, For the Figure 7, error bars and results of significance difference analysis should be presented. names of cassava cultivars mentioned in this table should be provided.
6, Numbering in subsections of results (3.1, 3.2, …) is incorrect, which should be 2.1, 2.2…
7, For the Materials and methods, experimentation site should be presented on a map, and genotype of plant samples should be clarified. Method for the statistical analysis should be described.
8, A conclusion section should be included.
Author Response
Reviewer 3
Comments and Suggestions for Authors
In this manuscript, Ciara Duff and colleagues examined how plant size, reproductive stage, and canopy structure influence trait variation in Bonellia nervosa during the dry season. I have following comments:
1, For the Abstract section, more results like thickness, water content, and stomatal density should be included. RESPONSE: We appreciate the suggestion. However, these traits did not show significant differences among the groups studied, as detailed in the main text. Given the strict word limit of the Abstract, we prioritized reporting traits with significant variation (e.g., SLA) and those most relevant to our main conclusions.
2, For the keywords, words appeared in the title should be removed from the list. RESPONSE: We thank the reviewer for spotting this inconsistency. We have removed keywords that appear in the article’s title.
3, For the introduction section, main conclusion and practical interests should be stated in the last paragraph. RESPONSE: We agree that the last paragraph of the Introduction should guide the reader toward the scope and significance of the study. However, to maintain the logical flow of the manuscript and avoid revealing key findings before presenting the supporting evidence, we have chosen to highlight the study objectives, hypotheses, and potential implications rather than stating the conclusions outright. This approach sets up the research questions while preserving the progression from evidence to conclusion in the Results and Discussion sections. We discuss the implications and practical interests of this research in the last paragraph.
4, For the Table 1, error values should be presented. RESPONSE: We thank the reviewer for this suggestion. However, eigenvalues and eigenvectors are deterministic quantities calculated directly from the data matrix decomposition and do not have associated error terms or standard errors in standard practice, unless the analysis has a different direction, such as resampling or bootstrapping, which is not the case here.
5, For the Figure 7, error bars and results of significance difference analysis should be presente. d. names of cassava cultivars mentioned in this table should be provided.
RESPONSE: The reviewer refers to Figure 8, since we had to shift the numbering of the figures. We appreciate the reviewer’s suggestion to include error bars and significance results in Figure 8. However, the rapid light curves for reproductive and non-reproductive plants are very close to each other, and no significant differences were found between the groups. Including error bars in the figure would visually clutter the presentation without adding interpretative value. We think it is unnecessary to repeat the same results in the figure legend since we clearly described the statistical results in the main text. The reference to Cassava is clearly a mistake by the reviewer.
6, Numbering in subsections of results (3.1, 3.2, …) is incorrect, which should be 2.1, 2.2…RESPONSE: We made this correction, as well as corrected the sequence of the figures.
7, For the Materials and methods, experimentation site should be presented on a map, and genotype of plant samples should be clarified. Method for the statistical analysis should be described. RESPONSE: We appreciate the reviewer’s suggestions. Santa Rosa National Park is one of the most extensively studied sites in tropical dry forest research, and its location is well documented in the literature. We therefore consider that providing the precise geographic coordinates, as currently stated, is sufficient for readers to locate the site without an additional map. Regarding genotype, our study takes an ecological and functional trait approach, and genetic variation among individuals was not part of our research scope or hypotheses. As such, genotype information is not relevant to our objectives or analyses.
8, A conclusion section should be included.
RESPONSE: As per the Plants author guidelines, a conclusion section is optional and should only be included if the authors believe the discussion is especially long or complex. We believe our findings are adequately addressed in the discussion section, and that it would be difficult to include a conclusion without being redundant.

Round 2
Reviewer 1 Report
Comments and Suggestions for Authors
Thank you for revising the manuscript following comments. I have no major or minor concerns about manuscript and can be accepted in its current form.
Reviewer 2 Report
Comments and Suggestions for Authors
Dear Authors.
I have reviewed in detail each of the responses. I consider that you have done an excellent scientific work and the manuscript has significantly improved its quality.
For this reason, I consider that in its current state it can be considered for publication.
Best regards.
Reviewer 3 Report
Comments and Suggestions for Authors
Authors have addressed my concerns in the revision.